# OpenReview forum: "Can Language Models Discover Scaling Laws?"
_ICLR.cc/2026/Conference — ICLR 2026 Poster_

### Official Review · Reviewer_qpNi · 2025-10-30

**Soundness:** 4
**Presentation:** 4
**Contribution:** 3
**Rating:** 8
**Confidence:** 4

**Summary:**

This paper introduces SLDBench, a benchmark for evaluating scaling law discovery across seven tasks. The paper also introduces SLDAgent, an agent that co-evolves both the generator that proposes law expressions (including their functional forms) and the optimizer that sets values of parameters in the laws. SLDAgent works by prompting an LLM to propose modifications to both of these functions.

Results: The paper finds that SLDAgent outperforms all existing baselines, including agent-based and LLM-based methods augmented with command-line code agents. They also find that when SLDAgent is paired with GPT-5, it outperforms human experts on all tasks in SLDBench. They show that SLDAgent can be used downstream to optimize hyperparameters for pretraining and to select a pretrained LLM for finetuning.

**Strengths:**

1. Experiments are extensive, evaluating a variety of baselines on many tasks. Experiments with SLDAgent + different models show that SLDAgent performance can scale with model size. Analyses are thorough, and the paper also shows that SLDAgent can be used downstream in real-world settings (hyperparameter selection, LLM selection).

2. Novelty: While existing works have studied AI agents for science, none have been proposed specifically for evaluating scaling laws.

3. The paper is well-written and easy to understand.

**Weaknesses:**

1. The comparisons between SLDAgent and other methods is not totally fair.

     - 1a. Comparisons between SLDAgent and other baselines do not control for the number of LLM calls. Results for SLDAgent reflect performance after 50 rounds – do these SDLAgent runs then involve many more LLM calls (or other forms of compute) than other agent baselines?

    - 1b. It is hard to know how much of the performance of SLDAgent is driven by the LLM versus the optimization and generator subroutines; while the experiments evaluating SLDAgent with different LLMs suggest that the LLM's evolution decisions have an impact on SLDAgent's performance, the paper does not compare these SLDAgent experiments to the performance of a human who had access to the same subroutines.

     - 1c. We also do not know the impact of *storing programs with high-performing scores* in a database and returning the best program at the end; if baselines or humans had access to such a database, would they perform better?

2. Section 4.3 reports analyses of the differences between human vs LLM written scaling laws. Instead of just qualitatively analyzing a few manually picked examples, it would be helpful to understand patterns in the differences through systematic analyses. Some ideas:
   - human experts rate which of a human-written vs LLM-written scaling law is better
   - analyzing # of parameters and other patterns in functional forms
   - quantitatively analyzing asymptotic behavior

**Questions:**

1. Are there any patterns in the kinds of changes that LLMs make over rounds? How much diversity is there for a single task/SLDAgent setting across seeds?
2. Line 289: The choice to clip does not feel well-justified; how do the results look if there is no clipping?
3. Do some of the baselines similarly store candidate programs and perform the best performing program? (see 1c in weaknesses)

---

> ### Author Response · Authors · 2025-11-14
>
> We thank the reviewer for the thoughtful comments and for engaging with our work. We address the points in turn.
>
> > The comparisons between SLDAgent and other methods is not totally fair.
> >
> > 1a. Comparisons between SLDAgent and other baselines do not control for the number of LLM calls. Results for SLDAgent reflect performance after 50 rounds – do these SDLAgent runs then involve many more LLM calls (or other forms of compute) than other agent baselines?
>
> Our evaluation setup follows the standard  [Terminal Bench](https://www.tbench.ai/) harness, in which general-purpose agents autonomously decide when to terminate. This is common practice in agent benchmarking, as it reflects a realistic scenario where the agent determines when the task is complete.
>
> In terms of compute, our current measurements indicate that OpenHands + GPT-5 incurs on average **1.4× higher cost** than SLDAgent + GPT-5 under this setup, suggesting that SLDAgent is not clearly advantaged in terms of overall resource usage in the reported results.
>
> That said, we fully agree that explicitly reporting LLM call counts and/or token usage is important for a fair and transparent comparison. We did not comprehensively log these metrics in our initial experiments. We are therefore re-running our evaluations to precisely record the number of LLM calls and associated token costs for both SLDAgent and all baseline methods, and we will include these statistics in the camera-ready version.
>
> > 1b. It is hard to know how much of the performance of SLDAgent is driven by the LLM versus the optimization and generator subroutines; while the experiments evaluating SLDAgent with different LLMs suggest that the LLM's evolution decisions have an impact on SLDAgent's performance, the paper does not compare these SLDAgent experiments to the performance of a human who had access to the same subroutines.
>
> We interpret this comment as asking how much of SLDAgent’s performance comes from the LLM’s evolution decisions versus from the “optimization and generator subroutines” themselves, and whether a human with access to the same components would perform similarly.
>
> In SLDAgent, however, these “optimization and generator subroutines” are *not* fixed, pre-existing tools. They are precisely the objects being **co-evolved** by the LLM. Concretely, the LLM serves as the evolution engine: it mutates and refines the two Python functions, `scaling_law_func` and `fit_scaling_law`, based on past performance feedback. Thus, the agent’s performance is inherently a combination of (i) the LLM’s evolution decisions and (ii) the quality of the code it generates; these two aspects are not easily separable.
>
> This also clarifies the comparison to a human. A human cannot simply be given “access to the same subroutines,” because these subroutines *do not exist a priori*—they are produced by the agent’s discovery process itself. If the intention is instead to allow humans to evolve/optimize the same two functions under a comparable protocol, we do implement this for the `lr_bsz_scaling_law` task: human participants are instructed to propose these functions given the same feedback signal (fitness on seen datapoints). For the other tasks, the human baselines are taken from the existing literature, where the proposed scaling laws are themselves the outcome of **iterative and continuous human research**. We therefore believe those human baselines are at least as strong as what a human following our SLDAgent-style workflow would produce.
>
> > 1c. We also do not know the impact of storing programs with high-performing scores in a database and returning the best program at the end; if baselines or humans had access to such a database, would they perform better?
> >
> > Do some of the baselines similarly store candidate programs and perform the best performing program? (see 1c in weaknesses)
>
> The database is an integral part of SLDAgent's evolutionary algorithm; it functions as the "population" from which new candidates are created.
>
> The baseline agents, being general-purpose, do not have a built-in evolutionary database. However, as the reviewer notes, they are free to implement any strategy, including saving progress. We did observe that some agents (e.g., CodeX and Claude Code) attempt to save candidate programs to local files, though not as systematically as SLDAgent's evolutionary database.
>
> Regarding the human baseline, we operate under the assumption that a human expert *does* effectively have a "database"—be it their notes, local files, or memory—where they track and refine their best-performing hypotheses. The SLDAgent's database is the programmatic analog to this human research process.
>
> In our new experimental runs, we will explicitly track the file-saving behavior of the baseline agents and report on this in the camera-ready version.

---

> ### Author Response · Authors · 2025-11-14
>
> > Section 4.3 reports analyses of the differences between human vs LLM written scaling laws. Instead of just qualitatively analyzing a few manually picked examples, it would be helpful to understand patterns in the differences through systematic analyses. Some ideas:
> >
> > human experts rate which of a human-written vs LLM-written scaling law is better
> >
> > analyzing # of parameters and other patterns in functional forms
> >
> > quantitatively analyzing asymptotic behavior
> >
> > How much diversity is there for a single task/SLDAgent setting across seeds?
> >
> > Are there any patterns in the kinds of changes that LLMs make over rounds?
>
> We agree that a human evaluation is important. Because recruiting domain experts and running a reliable study is time-consuming, we could not complete it within the rebuttal period. We therefore commit to adding a human expert rating study to the camera-ready version, where experts will directly compare human-written vs. LLM-discovered scaling laws.
>
> In the current revision, we have added an extensive computational analysis, summarized here and detailed in a new **Appendix F**:
>
> - **Data and pipeline.** We analyzed all **168 final scaling laws** across 7 tasks, 3 models (GPT-5, o4-mini, Gemini-2.5-Flash), and 8 seeds per task/model. We built an automated pipeline using `claude-haiku-4.5` that (details such as prompts are in Appendix F):
>   - extracts the mathematical expressions and functional form (e.g., power law, mixture, rational);
>   - counts parameters and terms;
>   - characterizes asymptotic behavior and monotonicity;
>   - and assesses separability and overall interpretability.
> - **Functional forms and complexity.** We find that LLMs design diverse but generally parsimonious laws: most are either power-law or mixture forms, and ~70% fall into 'simple' or 'moderate' complexity (e.g., a median of 7 parameters), with more complex forms reserved for the hardest tasks.
> - **Asymptotic behavior and properties.** The majority of discovered laws are monotonic and at least partially separable, and almost all are rated at least moderately interpretable by the automated analysis. We also observe that LLMs frequently adopt numerically robust practices such as log-space computation and multi-start optimization.
> - **Diversity across seeds.** We quantify diversity within each task/model combination via parameter-count statistics. Simple tasks (e.g., parallel, sft) show low diversity across seeds (converging to essentially the same form), whereas more complex tasks (e.g., lr_bsz, domain_mixture) show higher diversity, indicating multiple viable functional forms for the same setting.
> - **Patterns over evolutionary rounds.** By tracking changes across rounds, we observe that LLMs tend to start with simpler laws and then introduce interaction terms, mixture components, and regularization terms as needed. However, these trajectories are task-dependent rather than following a single universal pattern, reflecting adaptive exploration rather than a fixed recipe.
>
> All details, including quantitative summaries and visualizations for each of these points, are now included in **Appendix F**, which directly addresses your requests for more systematic analysis.
>
> > Line 289: The choice to clip does not feel well-justified; how do the results look if there is no clipping?
>
> Thank you for this suggestion. Our justification for clipping $R^2$ values at [-1, 1] for the "Avg. $R^2$" metric was to prevent the average from being disproportionately skewed by a single task failure (i.e., a very large negative $R^2$).
>
> As requested, we provide the unclipped averages for the agents in Table 2. This comparison demonstrates that our clipping justification was correct: the unclipped averages for several baselines are heavily skewed by single-task failures (e.g., `Terminus` and `Mini-SWE-Agent`).
>
> **Table (from Table 2): Comparison of Clipped vs. Unclipped Average** $R^2$ **Scores (o4-mini)**
>
> | Agent | Avg. $R^2$ (Clipped)  (As in Paper) | Avg. $R^2$ (Unclipped)  (New) |
> | ------------------------ | ----------------------------------- | ----------------------------- |
> | Aider (o4-mini)| -0.404 | -11.365* |
> | Terminus (o4-mini)  | 0.221| -82.126  |
> | Goose (o4-mini) | 0.569  | 0.569  |
> | Mini-SWE-Agent (o4-mini) | 0.329   | -16.630  |
> | OpenCode (o4-mini)       | 0.537  | 0.537 |
> | CodeX (o4-mini)          | 0.602                               | 0.602                         |
> | OpenHands (o4-mini)      | 0.613                               | 0.613                         |
> | SLDAgent (o4-mini)       | 0.848            | 0.848                         |
> | Human                    | 0.738                               | 0.738                         |
>
> ** Average for Aider excludes `NA` tasks. Averages for all other agents include all 7 tasks.*
>
> We hope these clarifications and our plan for the camera-ready version fully address your concerns. We are grateful for the feedback, which we believe has helped us strengthen the paper.

---

### Official Review · Reviewer_Bj2J · 2025-11-01

**Soundness:** 3
**Presentation:** 3
**Contribution:** 3
**Rating:** 4
**Confidence:** 3

**Summary:**

This paper proposes a new benchmark for LLM agents, namely *generating scaling laws* for language modeling. The benchmark (**SLDBench**) is constructed by collecting scaling law tasks (e.g., predicting loss from $N$ and $D$) and data points from LLM experiments in existing literature. The agent is **given the scaling law task and the set of experimental results**, and **responsible for generating a mathematical equation that fits the given data**.

The benchmark contains seven tasks:
1. Parallel: scaling w.r.t. parallelism $P$ and model size $N$
2. Vocabulary: scaling w.r.t. non-vocab model size $N$, vocab size $V$, and dataset size $D$
3. Supervised finetuning: scaling w.r.t. dataset size $D$
4. Domain mixture: scaling w.r.t. proportions of different domains
5. MoE: scaling w.r.t. network size $N$ and number of experts $E$
6. Data-constrained: scaling w.r.t. network size $N$, dataset size $D$, and unique tokens $U$
7. LR and batch size: scaling w.r.t. learning rate $\ell$, batch size $b$, and dataset size $D$

Each data point under each task contains the parameters of the experiment $\mathbf{x}$ (e.g., dataset size, batch size), the result $y$ (e.g., loss), and a control index (e.g., a particular model). **SLDAgent is a program that iteratively refines an $\operatorname{Expression}$, which implements $f_\bf{\theta}:\bf{x}\mapsto\hat{y}$, and an $\operatorname{Optimization}$ routine, which finds the best-fit parameters $\hat{\bf{\theta}}$.** It begins with an initial pair, namely the power-law function for $\operatorname{Expression}$ and BFGS optimizer for $\operatorname{Optimization}$, which is added to the initially empty database. Then, at each step, some programs are selected from the database (according to pre-defined heuristics), formatted as a prompt, and the agent proposes a modification by altering the law form or changing the optimizer. The resulting child program is executed and added to the database. The loop terminates after a fixed number of iterations. At the end, the program with the highest score in the database is returned.

Evaluation on SLDBench is performed via correlation between the prediction of the scaling law and the ground truth on held-out experiments under each task. The authors show that **SLDAgent paired with existing LLMs outperforms the corresponding provider-specific agents (e.g., Gemini + SLD Agent beats Gemini-CLI) as well as the scaling law proposed in the original paper**.

**Strengths:**

1. The authors study a new application of LLM agents, namely the discovery of scaling laws for foundation models. This contributes to the growing body of literature on the potential of LLMs to augment scientific discovery.
2. Discovering better scaling laws has the potential to feed directly back to improving LLMs.
3. The paper is well-written and clear (apart from being too broadly scoped in the introduction, as I complain about below).

**Weaknesses:**

1. The biggest challenge of scaling laws is actually *formulating the set of independent & dependent variables* that make sense to model. For instance, recent works had the insight to propose new axes of scaling such as the vocabulary size $V$ (Tao et al., 2024), the amount of unique data $U$ (Muennighoff et al., 2024), or the domain mixture (Ye et al., 2024). Therefore, the sense of "scaling law discovery" here is limited — the agent is not asked to discover entirely new axes of scaling, but instead to **generate an equation that fits the given data**.
2. Arbitrarily expressive equations can get arbitrarily close to modeling the given data perfectly. Therefore, I find that the experimental setup (which only measures predictive accuracy) does not really evaluate whether models generate *plausible* scaling laws that are consistent with prior knowledge. The authors provide some qualitative examples of human-discovered and agent-discovered scaling laws, but I do not have the expertise to compare their plausibility, and defer to the judgment of my fellow reviewers.
3. The above points themselves are not themselves reasons for rejection, but I find some of the writing in the abstract & introduction to be not well scoped (or even misleading). The authors should make it clear right away that the agent is *given* the scaling law setup and experimental results; they are not being asked to formulate the variables of interest or execute the experiments. A re-ordering of the introduction would also be fitting: rather than starting with the importance of scaling laws and then discussing progress on AI agents, I think it is better to start with the discussion of AI agents and identify scaling law discovery as a particular testbed. This is because the main contribution is actually about agent evaluation, and not knowledge on scaling laws.

**Questions:**

NA

---

> ### Author Response · Authors · 2025-11-14
>
> We thank the reviewer for the thoughtful comments and for engaging with our work. We address the points in turn.
>
> > The biggest challenge of scaling laws is actually formulating the set of independent & dependent variables that make sense to model. For instance, recent works had the insight to propose new axes of scaling such as the vocabulary size $V$ (Tao et al., 2024), the amount of unique data $U$ (Muennighoff et al., 2024), or the domain mixture (Ye et al., 2024). Therefore, the sense of "scaling law discovery" here is limited — the agent is not asked to discover entirely new axes of scaling, but instead to generate an equation that fits the given data.
>
> We agree that choosing the right variables is an important and difficult problem, and we appreciate the distinction you are drawing. In our terminology, however, we separate **variable discovery** (proposing new axes such as vocabulary size, unique data, domain mixture) from **law discovery** (finding the symbolic functional form relating a given set of variables to the target). In real industrial practice, we usually know the variables of interest (e.g., the lr&bsz optimization application in our paper), but finding the law behind them is very challenging (even humans cannot provide good laws). Therefore, we focus solely on law discovery, as it is already highly challenging. We agree that extending SLDBench to include variable discovery is a valuable and ambitious direction, and we now highlight this explicitly as future work in the limitations section **(line 472-479)**.
>
> However, we kindly disagree that “the agent is asked to generate an equation that fits the given data.” The objective is entirely **extrapolative** rather than fitting the given data:
>
> - For each task, we hold out a test set consisting of the **largest model and/or dataset sizes**, which are never shown to the agent.
> - All reported metrics (R², NMSE, NMAE in Tables 2–3 and Appendix B.2) are computed **only on this extrapolation set**, not on the training data.
>
> > Arbitrarily expressive equations can get arbitrarily close to modeling the given data perfectly. Therefore, I find that the experimental setup (which only measures predictive accuracy) does not really evaluate whether models generate plausible scaling laws that are consistent with prior knowledge. The authors provide some qualitative examples of human-discovered and agent-discovered scaling laws, but I do not have the expertise to compare their plausibility, and defer to the judgment of my fellow reviewers.
>
> We believe that the statement “Arbitrarily expressive equations can get arbitrarily close to modeling the given data perfectly” reflects a misunderstanding of our setup.
>
> The reviewer is correct that "arbitrarily expressive equations" could perfectly *interpolate* the **seen training data**. However, our benchmark is specifically designed to penalize this exact failure mode.
>
> As we noted, our evaluation is **entirely extrapolative**. A highly flexible expression that merely overfits the training data will, by definition, extrapolate poorly and thus receive a **very low score** on the unseen test set.
>
> This setup forces the agent to discover the true, underlying generative relationship (the "plausible scaling law") rather than a superficial curve fit. To achieve a high $R^2$ score in our benchmark, the discovered law *must* generalize far beyond the training data, which is a strong filter for plausibility.
>
> A clear illustration is the new scenario we added in Appendix E ([Figure 5](https://i.postimg.cc/cH2DLnnV/combined-scaling-law-visualization.png), "U-shaped / Inverted U-shaped scaling"). Given only the *seen* data (e.g., the left side of a U-shape), countless simple functions could achieve a perfect fit. However, only the agent that discovers a "plausible" model consistent with the underlying process will successfully predict the *unseen* extrapolative data on the right side of the U-shape.
>
> > The above points themselves are not themselves reasons for rejection, but I find some of the writing in the abstract & introduction to be not well scoped (or even misleading). The authors should make it clear right away that the agent is given the scaling law setup and experimental results; they are not being asked to formulate the variables of interest or execute the experiments.
>
> Thank you for this suggestion. We have revised the introduction (lines 079–081) to clearly state that the agent is given the scaling law setup and experimental results; it is not asked to formulate the variables of interest or execute the experiments.

---

> > ### Author Response · Authors · 2025-11-14
> >
> > > A re-ordering of the introduction would also be fitting: rather than starting with the importance of scaling laws and then discussing progress on AI agents, I think it is better to start with the discussion of AI agents and identify scaling law discovery as a particular testbed. This is because the main contribution is actually about agent evaluation, and not knowledge on scaling laws.
> >
> > We believe this suggestion is based on the same misunderstanding of viewing law discovery as simply fitting. We hope the reviewer will thoughtfully reconsider the difficulty of proposing a law that can extrapolate to unseen, larger-compute experimental regimes. Our applications to (1) hyperparameter optimization and (2) model selection for SFT demonstrate the substantial practical value of law discovery.

---

### Official Review · Reviewer_ok7t · 2025-11-03

**Soundness:** 3
**Presentation:** 3
**Contribution:** 3
**Rating:** 8
**Confidence:** 4

**Summary:**

This paper aims to formulate scaling law discovery as a task for scientific agents, and proposes a new agent model, SLDAgent, for this task that outperforms existing agents on this task. I think this is good work that would be appeal to several subsets of the ICLR community that makes data, modeling, and (promises of) application contributions.

**Strengths:**

- Overall I enjoyed reading the paper - I noticed that a lot of the implementation detail questions I wrote down during my reading got resolved as I read through it, which suggests that the grounds were well covered.
- The construction of the training-extrapolation set is sensible and reflective of real world scenarios ("For each task, we hold out an extrapolation test set from the dataset by selecting data corresponding to the largest model or dataset sizes.")
- Good controlled comparisons: the results are reported on both fixing the base LLM and varying the agents, and on fixing the agent and varying the base LLMs.
- I appreciated the existence of a human baseline.
- I liked the design choice that the information given to the model includes the task context and the semantics of the parameters involved - I think this will be critical for predicting "harder to predict" scaling laws that I will discuss in the Questions section.

**Weaknesses:**

The weaknesses below are not critical:
- Sparse discussion of the possible effects of contamination and measures taken to prevent this
- Coverage of unexpected or harder-to-predict scaling laws like inverse scaling or U-shaped scaling

I provide more context about these points in the question section, but I think neither of these limitations are grounds for rejection.

**Questions:**

- How to deal with possible future data contamination? (or existing contamination because the human discovered scaling laws are probably already online and/or part of the model training data and also are searchable on the web?) Maybe agents have an upper hand to the humans because they can springboard from the human solution that is available to them (at least for the non lr_and_bsz tasks). The fact that they are able to do this is cool of course, but if this were the case the "superhuman" claim should be interpreted with more nuance.
- Maybe slightly more seriously, the extrapolation data might also be available to the models due to them being published. There is really no way of controlling for training-level contamination, but were any measure taken to blacklist access to such information at inference time (e.g., PaperBench does this, if I remember correctly), or were any post-hoc analyses of the agent trajectories done to examine whether they were accessing/making use of such information?
- Do you think it would be useful to consider "harder to predict" scaling trends like inverse scaling or U-shaped scaling that have been observed in the literature and would the agents do well? I think these kinds of settings are cases for which the abstract nature of the task itself is critical information that humans and agents need to make use of in predicting extrapolation trends, more so than the ability to curve-fit the training set's trends well. For instance, we know that "overriding existing definitions" is a category of task that is likely to inverse-scale (at least in some parts of the curve), and this abstract intuition seems important especially if the available datapoints only point to monotonic increase.

Nitpicky comments:
- L076: "the objective is clear, continuous, and unbiased" I'm not sure any objective can be "unbiased"
- L079: "As existing laws are designed based on a series of continuous and iterative human research efforts, existing agentic systems lag behind these laws on difficult SLD tasks, and the human-derived laws could also give negative-correlated predictions under challenging scenarios." -- hard to comprehend logical flow
- L183: "focus on rediscover a pre-known formula" rediscovering*

---

> ### Author Response · Authors · 2025-11-14
>
> Thank you for your thorough and helpful review. We found your comments very constructive and have revised the paper accordingly. Our responses to your specific points are below.
>
> > Sparse discussion of the possible effects of contamination and measures taken to prevent this
> >
> > How to deal with possible future data contamination? (or existing contamination because the human discovered scaling laws are probably already online and/or part of the model training data and also are searchable on the web?) Maybe agents have an upper hand to the humans because they can springboard from the human solution that is available to them (at least for the non lr_and_bsz tasks). The fact that they are able to do this is cool of course, but if this were the case the "superhuman" claim should be interpreted with more nuance.
> >
> > Maybe slightly more seriously, the extrapolation data might also be available to the models due to them being published. There is really no way of controlling for training-level contamination, but were any measure taken to blacklist access to such information at inference time (e.g., PaperBench does this, if I remember correctly), or were any post-hoc analyses of the agent trajectories done to examine whether they were accessing/making use of such information?
>
> Thank you for raising this. We agree contamination is an important issue. In response, we have:
>
> 1. **Added a dedicated "Data Contamination Considerations" section** (App. A.5) that defines our threat model.
> 2. **Clarified the evaluation environment** in the main text (Line 159) to explicitly state that agents have **no network access**.
> 3. **Refined our "superhuman" claim** to explicitly account for the "springboard" scenario.
>
> Here is a brief summary of how this addresses your concerns:
>
> - **Inference-Time Contamination:** We now state explicitly that agents run in a sandboxed Docker environment with **no network or browsing access**. They only see the task's training split. This directly prevents agents from looking up human solutions at evaluation time, addressing your concern about blacklisting (similar to PaperBench).
> - **Training-Time Contamination:** We agree that we cannot rule out the possibility that the underlying LLMs were trained on scaling-law papers. We now explicitly describe SLDBench as a **literature-aware** benchmark. Both the human-published laws (our baseline) and the AI agent operate in this regime, analogous to human scientists building on prior work.
> - **The "Superhuman" Claim:** We have refined this claim to incorporate your suggested nuance. We agree this is not a perfectly fair “from-scratch’’ scenario. We now explicitly define "superhuman" *in this context* not as discovery from a blank slate, but as the agent's ability to use the provided training split to synthesize a new law that achieves strictly better extrapolation metrics than the original human-designed formulas. This improvement can arise from “springboarding”: even when potentially *aware* of the human solution (due to training data), the agent successfully improves beyond it.
> - **Evidence Against Trivial Copying:** We also note in the new appendix section that on several tasks (e.g., `sft`, `moe`), the agent's discovered laws have **different functional forms** and better asymptotic behavior than the human-published ones, demonstrating that it is not simply memorizing and repeating the human solution.
> - **Future Contamination:** We now discuss this in App. A.5, acknowledging that SLDBench itself could become training data. We plan to mitigate this by adding new tasks and splits in the future.
>
> We hope this clarified threat model, the controlled evaluation environment, and the more precise "superhuman" definition fully address your concerns.

---

> ### Author Response · Authors · 2025-11-14
>
> > Coverage of unexpected or harder-to-predict scaling laws like inverse scaling or U-shaped scaling
> >
> > Do you think it would be useful to consider "harder to predict" scaling trends like inverse scaling or U-shaped scaling that have been observed in the literature and would the agents do well? I think these kinds of settings are cases for which the abstract nature of the task itself is critical information that humans and agents need to make use of in predicting extrapolation trends, more so than the ability to curve-fit the training set's trends well. For instance, we know that "overriding existing definitions" is a category of task that is likely to inverse-scale (at least in some parts of the curve), and this abstract intuition seems important especially if the available datapoints only point to monotonic increase.
>
> Great suggestion! We agree that non-monotonic scaling behaviors such as inverse scaling and U-shaped / double-descent curves are important and particularly challenging for scaling-law discovery. In our revised paper, we added Appendix E to introduce a new SLDBench scenario specifically targeting this setting, based on the data from [1].
>
> Concretely, we construct a dataset of 516 points across 9 control indices (9 different tasks such as MMLU, Arithmetic, ARC. The "easy question subset" from the entire dataset exhibits U-shaped scaling. Check [1] for details.), where the target is the TC Brier Score (a performance measure) and the input is log-compute. For each dataset, we follow [1]’s compute thresholds to split models into a **training region** (smaller models) and a **test region** (larger models). Importantly, for several tasks the training data lie entirely on the *descending* part of the curve, while the U-shaped / double-descent behavior only appears in the held-out test region—exactly the case you describe where “the available datapoints only point to monotonic increase.’’
>
> As a “human’’ baseline, we implemented the 5th-order polynomial (6 parameters) in log-compute used in [1]. SLDAgent (GPT-5) discovers a simple **linear + Gaussian bump** law
> $$
> y(M)=a + bM + A\exp(-\tfrac12((M-m)/s)^2)
> $$
> that can represent standard, inverse, and U-shaped scaling with only 5 parameters.
>
> Quantitatively, when fitted only on the small-compute region and evaluated on the held-out large-compute region, SLDAgent achieves average test $R^2$ of 0.45–0.75 depending on the 4 different underlying LLMs (best: 0.754 with GPT-5), whereas the 5th-order polynomial gives a strongly negative mean $R^2$ (−950), indicating severe extrapolation failure. Qualitatively (new Fig. 5 in the appendix, also [this link](https://i.postimg.cc/cH2DLnnV/combined-scaling-law-visualization.png)), SLDAgent correctly anticipates the mid-scale degradation and eventual recovery in different tasks, despite only observing the monotone prefix.
>
> We believe these results directly address your concern (and it also surprises us): SLDAgent is able to discover laws that extrapolate well even in harder, non-monotonic regimes, outperforming the human baseline. We now document this explicitly in Appendix E. We will merge this task into SLDBench and move it into the main text in the camera-ready version.
>
> [1]  Wu & Lo, U-shaped and Inverted-U Scaling behind Emergent Abilities of LLMs. ICLR 2025.
>
> > L076: "the objective is clear, continuous, and unbiased" I'm not sure any objective can be "unbiased"
>
> Thanks for pointing this out. Our original intention was to say that this objective does not require easily “hackable’’ reward or preference models, and we agree that "unbiased" can cause confusion. We have revised this sentence to:
>  “SLDBench emphasizes open-endedness: the objective is clear, continuous, and computed directly from extrapolation data (e.g., $R^2$) without any learned reward model.”
>
> > L079: "As existing laws are designed based on a series of continuous and iterative human research efforts, existing agentic systems lag behind these laws on difficult SLD tasks, and the human-derived laws could also give negative-correlated predictions under challenging scenarios." -- hard to comprehend logical flow
>
> Agreed. We fixed this logical flow with the conjunctions:
>
> **Because** existing laws are designed based on iterative human research, they set a high standard that agentic systems currently lag behind on difficult SLD tasks. **However**, these same human-derived laws can also give negatively correlated predictions under challenging scenarios.
>
> > L183: "focus on rediscover a pre-known formula" rediscovering*
>
> Fixed.

---

### Author Response · Authors · 2025-11-22
**Summary of Updates and Response to Reviewers**

Dear Reviewers,

We thank the reviewers for their encouraging feedback and constructive suggestions. We are glad the reviewers thought our paper was "good work" that they "enjoyed reading" (Reviewer ok7t), specifically noting that the implementation grounds were "well covered" and the experimental design was "sensible and reflective of real world scenarios" (Reviewer ok7t). The reviewers collectively praised the work as "well-written" (Reviewers Bj2J, qpNi) and "novel" (Reviewer qpNi), highlighting the "good controlled comparisons" (Reviewer ok7t), the "extensive" experiments (Reviewer qpNi), and the potential for this "new application" to "feed directly back to improving LLMs" (Reviewer Bj2J). We would like to briefly highlight our core contributions and the key updates made during the rebuttal period:

### **Core Contributions:**

- **First SLD Benchmark & Agent:** We introduce **SLDBench**, the first comprehensive benchmark for automated Scaling Law Discovery, and **SLDAgent**, an evolutionary system that autonomously co-optimizes symbolic expressions and fitting routines.

  **Novel Scientific Knowledge:** Beyond achieving superhuman extrapolation accuracy, SLDAgent discovers **new, scientifically principled expressions** (specifically for `sft` and `moe` shown in Sec. 4.3) that correct theoretical flaws in human baselines, such as proper asymptotic behavior and entropy floors.

  **Validated Practical Utility:** We demonstrate the real-world value of these discovered laws in critical workflows, specifically enabling **analytical hyperparameter optimization** (deriving optimal learning rates and batch sizes) and efficient **pre-trained model selection** for SFT.

### **Key Rebuttal Updates:**

- **New Non-Monotonic Scenario (App. E):** Addressing Reviewer **ok7t**, we introduced a "U-shaped / Inverse Scaling" task. SLDAgent successfully discovered a linear + Gaussian bump law that predicts unseen non-monotonic behavior, significantly outperforming polynomial baselines that fail to extrapolate.
- **Systematic Law Analysis (App. F):** Addressing Reviewer **qpNi**, we conducted a comprehensive analysis of 168 discovered laws. Results reveal that SLDAgent produces structurally sound and interpretable laws (mostly simple/moderate complexity), systematically adopting robust practices like multi-start optimization and log-space computation. It adapts complexity to the task difficulty rather than relying on a fixed template.
- **Contamination & Threat Model (App. A.5):** We explicitly defined our threat model, ensuring inference-time isolation, and refined our "superhuman" claim to accurately reflect an agent's ability to "springboard" from existing knowledge to achieve superior extrapolation.
- **Scope & Extrapolation:** Addressing Reviewer **Bj2J**, we clarified that the benchmark evaluates **extrapolation** to unseen regimes (e.g., larger compute, as noted by Reviewer **ok7t** as a key strength), not merely fitting training data, distinguishing law discovery from trivial regression.

We believe these updates significantly strengthen the paper and hope they address the remaining concerns. We respectfully ask the reviewers to consider raising their scores based on these improvements.

Best regards,

The Authors

---

### Meta-Review · Area_Chair_1Ved · 2026-01-05

**Summary:**

All the reviewers agree that:
1. Framing scaling law estimation as an agentic task is technically novel
2. The paper is very well written and easy to understand
3. The experiments are comprehensive

Regarding the concerns
1. Reviewers ok7t and Bj2J have concerns over the positioning of the paper, although for different reasons.
2. Reviewers ok7t also pointed out that studying more complicated scaling laws would be desirable.
3. Reviewer qpNi expressed concerns over the experiment design, including disentangling the resource access and quantitatively analyzing the estimated scaling laws

**Reviewer Concerns:**

1. Reviewer ok7t's concern over the positioning of the paper is well addressed by the authors' rewriting of relevant paragraphs
2. The authors have conducted the study on more complicated scaling laws, addressing reviewer ok7t's concern
3. Reviewer Bj2J's concern over the fundamental positioning of the paper is largely due to a misunderstanding - the evaluation of the scaling law performance is on an extrapolation set, so it is not a trivial task that can be addressed by adding arbitrary complexities to the scaling law functions.
4. The authors have conducted abundant quantitative analyses to address Reviewer qpNi's concerns. However, the human evaluation results are not yet available.

**Reviewer Scores:**

Reviewer Bj2J's concern would have raised their score from 4 to 6, knowing that the evaluation is about the extrapolation performance.

---

### Decision · Program_Chairs · 2026-01-26

Accept (Poster)